# Oilseed Cakes: A Promising Source of Antioxidant, and Anti-Inflammatory Agents—Insights from *Lactuca sativa*

**DOI:** 10.3390/ijms252011077

**Published:** 2024-10-15

**Authors:** Mayye Majed, Amal A. Galala, Mohamed M. Amer, Dirk Selmar, Sara Abouzeid

**Affiliations:** 1Pharmacognosy Department, Faculty of Pharmacy, Mansoura University, Mansoura 35516, Egypt; mayye.majed@mans.edu.eg (M.M.); amal_galala@yahoo.com (A.A.G.); pharmamer_47@yahoo.com (M.M.A.); 2Pharmacognosy Department, Faculty of Pharmacy, Horus University in Egypt (HUE), New Damietta 34517, Egypt; 3Institute for Plant Biology, Technical University of Braunschweig, Mendelssohnsstr. 4, 38106 Braunschweig, Germany

**Keywords:** oilseed cake, lettuce, antioxidant, antimicrobial assay, cytotoxicity, anti-inflammatory

## Abstract

This study evaluated the antioxidant and antibacterial properties of methanolic extracts derived from oilseed cakes of *Lactuca sativa* (lettuce), *Nigella sativa* (black seed), *Eruca sativa* (rocket), and *Linum usitatissimum* (linseed). Lettuce methanolic extract showed the highest potential, so it was selected for further investigation. High-performance liquid chromatography (HPLC-DAD) analysis and bioassay-guided fractionation of lettuce seed cake extract led to the isolation of five compounds: 1,3-propanediol-2-amino-1-(3′,4′-methylenedioxyphenyl) (**1**), luteolin (**2**), luteolin-7-O-*β*-D-glucoside (**3**), apigenin-7-O-*β*-D-glucoside (**4**), and *β*-sitosterol 3-O-*β*-D-glucoside (**5**). Compound (**1**) was identified from *Lactuca* species for the first time, with high yield. The cytotoxic effects of the isolated compounds were tested on liver (HepG2) and breast (MCF-7) cancer cell lines, compared to normal cells (WI-38). Compounds (**2**), (**3**), and (**4**) exhibited strong activity in all assays, while compound (**1**) showed weak antioxidant, antimicrobial, and cytotoxic effects. The anti-inflammatory activity of lettuce seed cake extract and compound (**1**) was evaluated in vivo using a carrageenan-induced paw oedema model. Compound (**1**) and its combination with ibuprofen significantly reduced paw oedema, lowered inflammatory mediators (IL-1β, TNF-α, PGE2), and restored antioxidant enzyme activity. Additionally, compound (**1**) showed promising COX-1 and COX-2 inhibition in an in vitro enzymatic anti-inflammatory assay, with IC_50_ values of 17.31 ± 0.65 and 4.814 ± 0.24, respectively. Molecular docking revealed unique interactions of compound (**1**) with COX-1 and COX-2, suggesting the potential for targeted inhibition. These findings underscore the value of oilseed cakes as a source of bioactive compounds that merit further investigation.

## 1. Introduction

Oilseed crops are rich and well-known sources of nutrients and diverse biologically active compounds [1]. Cold-pressed seed oils are mostly utilized as dietary supplements and as ingredients in different skin care products. They are reliable sources of compounds that are medicinally and biologically active [2]. Worldwide oil plant production is expanding as oils extracted from seeds have become increasingly popular as food additives in recent years [3]. Moreover, oilseed meals/cakes, which are the remaining after the oil pressing process, are considered agro-industrial waste. Recently, they have been used in the production of biodiesel as well as animal feed [4]. Flaxseed cakes have undergone engine testing as a substitute feedstock for biodiesel, indicating that the future output of this oil cake might be anticipated to rise [5]. There are limited phytochemical and biological studies of seed cakes, although they are rich sources of carbohydrates, proteins, minerals, fibers, and certain lipids, as well as phytochemicals like phenolics (e.g., *p*-hydroxybenzoic, syringic, and *p*- coumaric acids) and flavonoids [6]. In this context, oilseed cakes are a very good example of agro-industrial waste that needs more attention in order to provide secondary metabolites with implementation in the medical and business sectors. Accordingly, it encouraged us to do further phytochemical and biological investigations of various medicinally important oilseed cakes (e.g., lettuce, flaxseed, black seed, and rocket seed meals).

Lettuce, black seed, rocket, and flaxseed seeds contain different compounds under the classes of flavonoids and phenolic acids, which are known and reported for their antioxidant activity [7,8,9]. Moreover, lettuce and rocket seeds were reported to have analgesic and anti-inflammatory activities [9,10]. Additionally, black seed was stated for various biological activities such as antioxidant, antibacterial, and neuroprotective effects [11]. In vitro screening of flaxseed demonstrated potentials for antibacterial, antioxidant, anti-diabetic, and anti-inflammatory properties [12,13]. Concerns about health and safety have led to the use of natural antioxidants in the food sector as a substitute for synthetic ones (e.g., phenolic compounds). Natural phenolic compounds are anticipated to have a steady increase in interest since many of these molecules have been shown to exhibit a wide range of biological significance, including antimutagenic, antitumor, having protective properties against inflammations, platelet aggregation inhibitory, neuroprotective, antimicrobial, and many more [14]. Consequently, adding these substances to everyday products would stabilize food as well as cosmetic products and provide significant health benefits to customers.

In this study, the antibacterial and antioxidant activities of four seed cake extracts (e.g., lettuce, flaxseed, black seed, and rocket) were examined. Moreover, further chromatographic and anti-inflammatory investigations were performed on the seed cake with the highest bioactivity.

## 2. Results and Discussion

Agro-industrial wastes are a hidden treasure of valuable bioactive compounds that require more investigation. Oilseed cakes, which are leftovers from the oil extraction of many seeds and fruits, are a very good example of plant waste that needs more attention as a source of bioactive compounds. They occur as by-products at the end of the pressing process, and they are rich in some major and minor components. In addition, oilseed cakes contain a high percentage of nutrients besides antioxidant and antimicrobial bioactivity and thus could be used as nutraceuticals and functional foods [15,16]. In our study, we focused on antioxidant and antimicrobial screening of the different seed cakes/meals. Furthermore, the highest bioactive seed cake was subjected to further bio-guided isolation of active constituents.

### 2.1. Anti-Oxidant and Antimicrobial Screening of Total Methanolic Extracts from Different Oilseed Cakes

Lettuce, black seed, rocket, and flaxseed seed contain different compounds under the class flavonoids and phenolic acids, which are known and reported for their antioxidant and antimicrobial activity [7,8,9]. The antimicrobial and antioxidant properties of the seed cakes’ methanolic extract were assessed at 1 mg/mL. Lettuce seed cake extract showed the highest anti-oxidant activity with 36.1% inhibition, followed by black seed 17.3%, rocket 14%, and flaxseed 10.5%, respectively, compared to ascorbic acid standard 88.1% (Appendix A). Whilst the only lettuce seed cake extracts showed antimicrobial activities against *S. aureus*, *E. coli*, and *C. albicans* (Appendix A). The highest inhibitory activity of lettuce was observed against *C. albicans*, followed by *S. aureus* and *E. coli*, with a zone of inhibitions at 6.83 ± 0.29, 3.75 ± 0.25, and 2.8 ± 0.29 mm, respectively. In addition, flaxseed and rocket seed cakes showed low antimicrobial activities only against *S. aureus* (1.75 ± 0.25, 1.8 ± 0.25 mm) and *C. albicans* (4.75 ± 0.25, 5.85 ± 0.13 mm), respectively. Black seed cake could not inhibit the growth of all tested microorganisms (Appendix A). It was noticeable that lettuce extract was the only extract among the tested ones to give broad-spectrum activity against *S. aureus*, *E. coli*, and *C. albicans*, while other extracts gave activity against only one organism or two compared to ampicillin and clotrimazole standards. Accordingly, oilseed cakes represent a promising agro-industrial waste with considerable biological activities. It has been reported that oilseed cakes from various plants, such as canola, sunflower, and mustard, exhibit antioxidant and antimicrobial properties, likely due to their phenolic and flavonoid content [17]. Furthermore, *Jatropha curcas* seed cake could be combined with commercial antibiotics to enhance their effectiveness, offering a potential solution for treating clinical infections, including those caused by multidrug-resistant bacteria [18].

### 2.2. Antioxidant and Antimicrobial Activities of Different Fractions of Lettuce Seed Cake Extract

Based on the previous activity results, the most promising cake was selected for further biological and phytochemical investigation. As lettuce methanolic extract showed dual activity, and due to the scarcity of previous work on this seed and its waste, it was a valuable candidate for further phytochemical and biological investigations. Lettuce seed cake methanolic extract was subjected to fractionation, which afforded petroleum ether, methylene chloride, and ethyl acetate fractions. Ethyl acetate fraction showed the highest antioxidant activity, followed by the methylene chloride fraction with IC_50_ 27.13 ± 0.15 and 41.93 ± 0.21, respectively, compared to standard ascorbic acid 29.47 ± 0.17 (Figure 1). Moreover, methylene chloride and ethyl acetate fractions showed promising antimicrobial inhibition activity against the tested microorganisms with comparison with antibacterial ampicillin and antifungal clotrimazole standards (Appendix A). The comparison with the standard revealed that the highest activity in terms of percent activity index was observed for the methylene chloride fraction against *S. aureus* (81.6), followed by *C. albicans* (65.4), and *E. coli* (36.26) (Appendix A).

### 2.3. Bio-Guided Isolation of Active Constituents from L. sativa Seedcake Active Fractions

Using reversed-phase HPLC fitted with a diode array detector, the relevant extract was fractionated to determine the phytochemical components of lettuce seed cake extract that are in charge of the antioxidant and antibacterial actions. It was observed that luteolin glucoside represents the major compound (Figure 2). This could be easily deduced by comparing the retention time and UV spectrum of standard luteolin glucoside (Appendix A). In addition, the less polar compound that was eluted at Rt 26 mint has the same UV spectrum as luteolin glucoside (Appendix A). This compound was tentatively identified as an aglycon part of luteolin glucoside, e.g., luteolin. However, reliable identification of these compounds requires much more information, and this cannot be achieved before the isolation and spectroscopic identification by NMR techniques. In the same manner, the identification of the other peaks displayed in Figure 2 also required much more information.

Accordingly, methylene chloride and ethyl acetate fractions were subjected to normal silica column chromatography, which afforded five compounds (Figure 3). The compounds were identified depending on NMR data as 1,3-propanediol-2-amino-1-(3′,4′-methylenedioxyphenyl) (**1**) [19], luteolin (**2**) [20], luteolin 7-O-*β*-D-glucoside (**3**) [21], apigenin 7-O-*β*-D-glucoside (**4**) [22], and *β*-sitosterol 3-O-*β*-D-glucoside (**5**). Compound (**5**) was characterized upon comparison with an authentic standard sample. For more details of structure elucidation, see the Appendix A. Luteolin (**2**) and luteolin 7-O-*β*-D glucoside (**3**) were previously reported to be isolated from lettuce [23], however, up to our knowledge, this is the first time apigenin 7-O-*β*-D glucoside (**4**) has been reported from *L. sativa* seed cake. It is worth mentioning that compound (**1**) is the first time to be isolated from *Lactuca* species, the second time from the Asteraceae family [19], and the third time from nature [24].

### 2.4. Biological Evaluation of the Isolated Compounds

The isolated compounds were evaluated for their antioxidant, antimicrobial, and cytotoxicity. Compounds (**3**) and (**4**) showed strong antioxidant activity of IC_50_ 35.18 ± 0.18 and 51.27 ± 0.25, respectively. Meanwhile, compound (**2**) showed moderate activity with an IC_50_ 60.49 ± 0.29. In contrast, compounds (**1**) and (**5**) showed relatively weaker antioxidant activity compared to the ascorbic acid standard (Figure 4). In addition, compounds (**2**) and (**3**) exhibited promising antimicrobial activity throughout the tested compounds, with the highest percent activity index against *S. aureus* (71.7, 61.9), followed by *P. aeruginosa* (56.6, 51.5), *C. albicans* (40.5, 36.7), and *E. coli* (37.8, 33.8), respectively (Table 1). In addition, compounds (**4**) and (**5**) also possess antimicrobial activity but with a much lower percentage, especially against *S. aureus*, with percent activity 50 and 49.6, respectively. On the other hand, compound (**1**) showed weak anti-microbial activity compared to ciprofloxacin and clotrimazole standards (Table 1). The potential antioxidant and antimicrobial activity of methylene chloride and ethyl acetate fractions could be attributed to the content of compounds (**2**) and (**3**) in these active fractions. There are previous studies that reported both the antioxidant and antimicrobial activities of luteolin and its glucoside [25,26,27]. Accordingly, they are the major compounds responsible for the antioxidant and antimicrobial capacity of *L. sativa* seed cake.

Moreover, compounds (**2**) and (**3**) demonstrated potent cytotoxic action against the cancer cell lines (HepG-2) and (MCF-7) with IC_50_ (19.50 ± 1.3, 11.63 ± 0.9), and (24.83 ± 1.9, 17.79 ± 1.3), respectively. Compared to compounds (**5**) and (**1**), which had minimal anticancer action, compound (**4**) demonstrated moderate efficacy with IC_50_ (43.35 ± 2.5, 39.79 ± 2.2) (Table 2). To investigate the safety and selectivity of the isolated compounds, the normal lung fibroblast cell line (WI-38) was used in the assessment process. In comparison to ordinary doxorubicin, which had selectivity indices of 1.5 for (HepG2) and 1.6 for (MCF-7) cell lines, respectively, compounds (**2**) and (**3**) showed greater selectivity indices (3, 2) for the (HepG2) cell line and (5.1, 2.8) for the (MCF-7) cell line, indicating higher safety profiles as anticancer drugs. On HepG2 and MCF-7 cell lines, compounds (**2**) and (**3**) exhibited cytotoxic action while maintaining a reasonable level of safety. There are also previous studies that reported the anticancer potential of the bioactive molecule luteolin and its analogs [28]. On the other hand, compound (**1**) cytotoxic power was very weak and almost nontoxic, which may lead to the possibility of anti-inflammatory potential. Compound (**1**) was reported before to have moderate inhibitory activities against SARS-CoV-2 main protease Mpro or 3CLpro [19], also showed strong inhibitory against IL-6 production in TNF-a-stimulated MG-63 cells [24], which supports our vision that compound (**1**) might be a potential anti-inflammatory agent.

### 2.5. Anti-Inflammatory Activity of Lettuce Seed Cake Extract and Isolated Compound *(**1**)* in Carrageenan-Induced Paw Oedema In Vivo

The carrageenan-induced paw edema test is widely used to assess the anti-inflammatory effects of new pharmaceutical agents [29,30,31]. In this study, the subcutaneous injection of carrageenan in rats led to a time-dependent increase in paw edema (Table 3). The inhibitory effects of different treatments are displayed in Table 3 and Figure 5 and Appendix A. Results showed that an oral dose of lettuce seed cake extract (2 g/kg) significantly reduced paw edema size (*p* < 0.05) after 4 h. Compound (**1**) at doses of 40 mg/kg caused significant inhibition of paw edema (*p* < 0.001) starting from the 3rd hour after carrageenan administration. Moreover, the combination of ibuprofen (10 mg/kg) with compound (**1**) (20 mg/kg) further reduced paw edema (*p* < 0.001) after 5 h, showing more promising results than the standard drug.

Histopathological analysis revealed that skin sections from Group 1 (negative control group) showed normal epidermis and dermis, including hair follicles, sebaceous glands, and collagen bundles, with no signs of inflammation. Microscopic images at higher magnification (Figure 5) displayed very few leukocytes in the dermis. In contrast, Group 2 (carrageenan group) exhibited severe dermal inflammation and significant leukocyte infiltration (thin black arrow). Treatment with lettuce seed cake extract in Group 4 led to a slight reduction in dermal inflammation. An oral dose of compound (**1**) at 20 mg/kg resulted in a moderate decrease in both inflammation and leukocyte infiltration, while Group 6, treated with compound (**1**) at 40 mg/kg, showed a substantial reduction in dermal inflammation. Group 3 (standard treatment) showed mild dermal inflammation, whereas the combination of compound (**1**) with standard ibuprofen (Group 7) demonstrated a synergistic effect, with no signs of inflammation and very few leukocytes (Figure 5).

#### 2.5.1. Impact on Enzymatic Anti-Inflammatory Activity

The effects of lettuce seed cake extract and compound (**1**) on the levels of inflammatory mediators, including IL-1β, TNF-α, and PGE2, in carrageenan-induced paw edema in rats are shown in Figure 6. Treatment with compound (**1**) resulted in a significant reduction in IL-1β and TNF-α levels compared to the vehicle control group at doses of 20 and 40 mg/kg (* *p* < 0.05, ** *p* < 0.01). Notably, the combination of ibuprofen (10 mg/kg) with compound (**1**) (20 mg/kg) yielded particularly promising results, significantly reducing IL-1β, TNF-α, and PGE2 levels (*** *p* < 0.001).

#### 2.5.2. Impact on Enzymatic Antioxidant Status

Carrageenan-induced local inflammation has been linked to the generation of reactive oxygen species (ROS) and is a significant contributor to oxidative stress. The results for SOD, CAT, and GSH levels in the paw edema tissue across different test groups are summarized in Table 4. Treatment with compound (**1**) at 40 mg/kg restored SOD activity by 71.43%, CAT activity by 72.08%, and GPx activity by 73.17% compared to the control group. The combination of ibuprofen (10 mg/kg) and compound (**1**) (20 mg/kg) showed even greater protective effects, with SOD, CAT, and GSH activities increasing by 85.08%, 84.28%, and 84.23%, respectively. In comparison, ibuprofen alone provided protection of 75.17%, 83.39%, and 78.74%, respectively. These findings suggest that both lettuce extract and compound (**1**) enhance antioxidant enzyme activities, potentially by bolstering the cellular antioxidant defense mechanisms. This indicates that they may offer protective benefits by stimulating the expression and activity of antioxidant enzymes during the inflammatory response.

The phytochemical analysis of lettuce seed cake suggests that the compound 1,3-propanediol-2-amino-1-(3′,4′-methylenedioxyphenyl) (compound **1**) is primarily responsible for its anti-inflammatory activity. Previous studies have shown that compound **1** effectively inhibits IL-6 production in TNF-α-stimulated MG-63 cells [23]. Additionally, flavonoids such as luteolin and luteolin-7-O-*β*-D-glucoside are well recognized for their anti-inflammatory properties [32,33]. As a result, *L. sativa* seed cake extract emerges as a promising candidate for anti-inflammatory applications. This study lays the foundation for further investigation into the mechanism of action of compound **1** and the potential uses of lettuce seed cake extract in the food and cosmetic industries.

### 2.6. In Vitro COX-1 and COX-2 Inhibition Assay and Molecular Docking for Compound *(**1**)*

Compound (**1**) demonstrated promising activity against the COX-1 and 2 inhibition assay in comparison to the ibuprofen standard. Additionally, Figure 7 shows that it was more active against COX-2 than COX-1, with IC_50_ values of 4.814 ± 0.24 and 17.31 ± 0.65, respectively. The results were supported by docking as unique interaction patterns of compound (**1**) with COX-2 and COX-1 (Figure 8).

The results of molecular docking and binding affinities between compound (**1**) and COX-1 and COX-2 proteins are shown in Figure 8 and Table 5. Red indicates alpha helices in the protein’s tthree-dimensional structure, green shows sheets, and restricted lines show the chemical bonds that have been established between the drug and COX.

In comparison to ibuprofen, compound (**1**) shows stronger binding to both COX-1 and COX-2, according to the binding affinity results, which are displayed in Table 5. Compared to ibuprofen (−5.4 kcal/mol for COX-1 and −4.8 kcal/mol for COX-2), the tested compound’s ΔG values (−6.6 kcal/mol for COX-1 and −5.4 kcal/mol for COX-2) indicate a higher binding affinity and possibly a larger inhibitory impact. This enhanced binding profile might point to a more effective anti-inflammatory effect of the investigated substance.

The 2D and 3D visualizations highlight the key residues involved in the interactions and provide a visual confirmation of the tabulated data (Figure 8). For COX-1, ASN59 and ASN77 are significant residues in the binding site that the tested compound makes hydrogen bonds with. The interaction’s binding affinity and specificity are greatly influenced by these hydrogen bonds. To further stabilize the binding, there are hydrophobic interactions between PHE81 and the vicinity of GLY55 and GLY56. While the tested chemical exhibited a robust hydrogen bonding network, ibuprofen’s interactions with COX-1 are largely hydrophobic.

Even more noticeable variations can be seen in the interactions with COX-2 (Figure 8). The evaluated substance establishes many hydrogen bonds with GLN195 and SER197, residues that are essential for the selectivity and efficacy of COX-2 inhibitors. Together with the hydrophobic contacts between PRO166 and LEU136, these hydrogen bonds point to a potent and distinct binding mechanism. However, in the COX-2 binding region, ibuprofen only exhibits hydrophobic interactions with ALA2 and PRO166, suggesting a potentially less ideal binding configuration. These differences in binding patterns and affinities have several implications for the biological activity of the investigated substance in comparison to ibuprofen. Compound **1** may have more potency as a COX inhibitor due to its stronger and more frequent interactions with both enzymes. The decreased binding energies (ΔG values) found in the docking studies provide support for this.

More research on the molecule may reveal its greater COX-2 selectivity, which could lead to better gastrointestinal tolerability when compared to non-selective NSAIDs such as ibuprofen. In this context, it is especially interesting that important residues like SER197 are involved in COX-2 binding. It is possible that the tested chemical will spend a longer amount of time at the active site of the enzyme due to its more stable binding mechanism and large hydrogen bonding network. This might result in a longer duration of action than ibuprofen, which would be beneficial for sustaining therapeutic benefits and dose frequency.

Moreover, the presence of crucial residues such as GLN195 and SER197 in COX-2 and ASN59 and ASN77 in COX-1 suggests that the investigated substance may successfully obstruct the entry of arachidonic acid, the natural substrate, or disrupt the catalytic functions of these enzymes. Therefore, while talking about the ADMET profile, ibuprofen has a Log *p* value of 3.687, which indicates good lipophilicity, which contributes to its ability to cross biological membranes. The high protein binding (PPB) of 94.37% suggests that ibuprofen circulates in the bloodstream primarily bound to plasma proteins. This characteristic can influence its distribution and half-life in the body. The low fraction unbound (Fu) of 3.65% further supports this high protein binding nature. The tested compound (**1**), in contrast, demonstrates different pharmacokinetic properties. With a lower Log *p* of −0.763, it is more hydrophilic than ibuprofen, which may affect its ability to cross cell membranes. Its protein binding is considerably lower at 31.56%, with a higher fraction unbound (72.23%), suggesting that it may have a different distribution profile in the body compared to ibuprofen (Appendix A).

Both compounds show low blood-brain barrier (BBB) penetration probabilities (0.463 for ibuprofen and 0.326 for the tested compound), indicating they may not readily enter the central nervous system. This could be beneficial for reducing potential neurological side effects but might limit their efficacy for CNS-related conditions. Therefore, further investigation of semi-synthetic derivatives could be suggested to produce derivatives with better profiles.

The information also includes the interactions of the compounds with cytochrome P450 enzymes, which are critical for drug metabolism. The information shows that ibuprofen is more likely than the tested chemical (0.171) to be a substrate for CYP2C9 (0.982) (Appendix A). This is in line with the enzyme’s documented function in ibuprofen metabolism. Being aware of this helps you prepare for potential drug interactions. According to toxicity predictions, there is a minimal likelihood of either chemical having serious negative effects. The studied compound’s hERG inhibition probability is 0.07, compared to 0.018 for ibuprofen, suggesting a low risk of cardiotoxicity. Important safety factors, such as genotoxicity (Ames test) and carcinogenicity, both exhibit low probabilities for these substances.

## 3. Conclusions

Oil seed cakes are hidden treasures for a variety of biologically active substances. The findings revealed that *L. sativa* seed cake extract contains various bioactive substances with potential antibacterial, antioxidant, and anti-inflammatory properties. Compound (**1**), both on its own and in combination with ibuprofen, significantly reduced the production of TNF-α, IL-1β, and IL-6, showed selective inhibition of COX-2, and decreased carrageenan-induced paw oedema in rats. Consequently, *L. sativa* seed cake may serve as a valuable natural resource for developing novel nutraceuticals, functional foods, and applications in the pharmaceutical and food industries.

Ultimately, oilseed cakes represent a promising agro-industrial waste with considerable potential due to their rich phytochemical content, including phenolics and flavonoids [17]. Despite their current use in biodiesel production and animal feed [34], further research is needed to fully explore their phytochemical and biological properties. This study showed the importance of oilseed cakes as a valuable resource for secondary metabolites, which could have significant applications in the medical and commercial sectors. As interest in sustainable practices grows, more attention should be given to utilizing these by-products for their broader economic and environmental benefits.

## 4. Materials and Methods

### 4.1. General Experimental

The HPLC was performed in the Institute for Plant Biology, TU Braunschweig, using a Young Lin quaternary pump, vacuum degasser, column oven (40 °C), diode array detector (YL9160 PDA), and a Midas Spark Holland autosampler. UV spectra (λ max) were measured in the central laboratory of the Faculty of Pharmaceutical Science at Mansoura University using spectroscopic methanol with an ultraviolet-visible spectrophotometer (Shimadzu 1601 PC, version TCC-240A, Kyoto, Japan). The NMR unit of Mansoura University’s Faculty of Pharmacy obtained nuclear magnetic resonance spectra (^1^H-NMR, ^13^C-NMR, DEPT-Q, APT, HSQC, and HMBC) using a Bruker DRX 600 NMR spectrometer (600 and 150 MHz for ^1^H and ^13^C-NMR, respectively) and with the Bruker Corporation Avance III 400 spectrometer, which measures ^1^H and ^13^C-NMR at 400 and 100 MHz, respectively. The types of solvents that are used are CDCl_3_, CD_3_OD, and DMSO-d6. For normal phase chromatography, silica gel G 60–230 mesh (Merck, Darmstadt, Germany) was packed using a wet or dry method in the specified solvent. The purchase of Normal Human lung fibroblast (WI-38), hepatic cancer cell lines (HePG-2), and mammary gland (MCF-7) cell lines from ATCC was made possible by the Holding firm for biological goods and vaccines (VACSERA), which is based in Cairo, Egypt.

The levels of inflammatory mediators (IL-1β, TNF-*α*, and PGE2) in rat paw tissue homogenate were measured using the following Kits: Rat TNF-α (Tumor Necrosis Factor Alpha) ELISA Kit from Elabscience^®^ Company, Houston, TX, USA, Interleukin 1 Beta (IL-1b) Organism Species: Rattus norvegicus (Rat) from Cloud Clone Crop Company and Rat PGE2 (Prostaglandin E2) ELISA Kit from Fine test^®^ company Hubei, China. In addition, oxidative stress parameters in rat paw tissue homogenate were measured using superoxide dismutase, glutathione peroxidase, and catalase Kits from the Biodiagnostic company, Birmingham, UK.

### 4.2. Chemicals

All solvents used were of HPLC grade and supplied by Fisher Scientific. Luteolin-7-glucoside standard was purchased from Carl Roth (Karlsruhe, Germany). The solvents used for extraction, chromatographic separation, and crystallization (i.e., petroleum ether, methylene chloride, ethyl acetate, and methanol) were of reagent grade, obtained from EL-Nasr Company for pharmaceutical chemicals, Egypt. ABTS (Azino-bis-(3-ethyl benzthiazoline-6-sulfonic acid), COX-1, COX-2, RPMI-1640 medium, MTT, DMSO, CMC, doxorubicin, ciprofloxacin, and clotrimazole were purchased from Sigma Co., St. Louis, MO, USA. Ascorbic acid (Cevarol^®^) and ibuprofen (Brofen) tablets were obtained from Memphis Pharmaceutical Co., Cairo, Egypt, and abbott pharmaceuticals, Abbott Park, IL, USA, respectively.

### 4.3. Plant Material

*Lactuca sativa* (lettuce), *Nigella sativa* (black cumin), *Eruca sativa* (rocket), and *Linum usitatissimum* (flaxseed) seed cakes were obtained from Albadawia Oil Factory, Ferdos City, Mansoura, Dakahliya, Egypt, 5 August 2022.

### 4.4. Extraction of Seed Oil Cakes Derived from Different Plant Species

For preliminary screening to obtain dried methanolic extracts from various oilseed cakes, 200 g each of lettuce, black seed, rockets, and flaxseed seed cakes were extracted three times by overnight maceration using methanol (3× 400 mL each). The methanolic extracts were separated from the crude powders through centrifugation and filtration using Whatman filter paper. The extracts were then concentrated under reduced pressure at 45 °C using a rotary evaporator. The resulting extracts were dried in a desiccator over anhydrous calcium chloride until they reached a constant weight, then stored at room temperature for biological assays and further chromatographic analysis.

### 4.5. HPLC Analysis of Lettuce Seed Cake Defatted Extract

HPLC separation was performed using a Nucleosil RP-C18 column (5 μm particle size, L × I.D. 25 cm × 3.2 mm). A binary gradient was applied, starting with 85% A (aqueous trifluoroacetic acid 0.05%) and 15% B (acetonitrile). After 5 min, the ratio was adjusted to 80% A and 20% B, followed by the following gradient: 15 min: 70% A, 30% B; 25 min: 20% A, 80% B; 30 min: 20% A, 80% B; 35 min: 10% A, 90% B; 37 min: 85% A, 15% B, and held until 47 min. The flow rate was set at 1 mL/min, with an injection volume of 20 μL. Detection was performed using a photodiode array (PDA) detector at 254 nm and 350 nm.

### 4.6. Bio-Guided Isolation of Active Constituents from Lactuca sativa Seed Cake Active Fractions

For chromatographic investigation of lettuce seed cake, 5 kg of the dried cake was extracted three times with methanol (10 L each) through overnight maceration in siphon extraction jars. The methanolic extracts were separated from the crude powders and concentrated under reduced pressure at 45 °C using a rotary evaporator. The resulting dried extract of 760 g of residue was suspended in water for fractionation using petroleum ether, methylene chloride, and ethyl acetate. The weights of the resulting fractions were as follows: 590 g for the petroleum ether extract (extremely thick and oily), 11 g for the methylene chloride extract, and 14 g for the ethyl acetate extract.

Column chromatography of regular silica gel (3.5 cm, 300 g) packed with methylene chloride solvent was applied to the methylene chloride fraction extract, eluting it with methylene chloride-methanol (95:5), and progressively raising the polarity of the eluent by 5%. Utilizing thin-layer chromatography, the resulting fractions’ homogeneity was assessed. Subfractions were obtained by combining similar fractions. Subfraction was further purified by crystallization from methanol and eluted with various systems to yield compound **1** (560 mg) with an R_f_ of 0.2 when developed on GF_254_ precoated silica gel plate using solvent system methylene chloride absolute (100%), and compound **2** (8 mg) with an R_f_ value of 0.60 when 10% methanol in methylene chloride was used. Subfraction 3 was chromatographed on silica to provide compound **5** (18 mg) R_f_ 0.26 after being eluted with methylene chloride-ethyl acetate (70:30).

Normal silica gel column chromatography packed with methylene chloride (3.5 cm, 350 g) was applied to the extract of ethyl acetate. Methanol was added once elution was completed using methylene chloride-ethyl acetate (95:5) and a 5% increase in eluent polarity up to 100% ethyl acetate. Thin-layer chromatography was employed as a monitoring tool to monitor the collected fractions’ homogeneity. By combining similar fractions, compounds **3** and **4** were produced. Chromatographic analysis of compound **3** (15 mg) on TLC-precoated silica gel plates GF_254_ using the solvent system 20% methanol in methylene chloride revealed an R*_f_* value of 0.32 and compound **4** (6 mg) as a yellow amorphous powder, which was further purified by reprecipitation with an R_f_ value of 0.46 using 15% methanol in the ethyl acetate system on TLC.

### 4.7. Biological Screening

#### 4.7.1. Antioxidant Screening (ABTS Assay)

The experiment was performed in triplicate according to Zaky et al. [35]. The absorbance of control of the resulting green-blue solution (ABTS* radical solution) was recorded at λ max 734 nm.

#### 4.7.2. Antimicrobial Screening

The procedure was performed according to Stylianakis et al. [36]. A comparison of the extract’s zone of inhibition to that of a reference antibiotic’s activity index was determined using the following equation:Activity index = (the inhibition zone of extract/the inhibition zone of the standard) × 100

#### 4.7.3. Cytotoxic Activity (MTT Assay)

Using the MTT test, the inhibitory effects of substances on cell growth were ascertained using the cell lines [37] compared to normal lung fibroblast cells (WI-38). The selectivity index was calculated via the Selectivity index (SI) equation: SI equals IC_50_ normal/IC_50_ cancer. IC_50_ normal refers to the concentration of the investigated substance that killed 50% of normal cells, whereas IC_50_ malignancy denotes the concentration that killed 50% of malignant cells [38].

#### 4.7.4. Anti-Inflammatory Test: Carrageenan-Induced Paw Oedema

Adult albino rats 180–220 g of both sexes used in the investigation were acquired from the Mansoura University Faculty of Pharmacy’s Animal House. Standard laboratory diet and water were provided, and they were housed in cages maintained at a constant temperature of 22 ± 2 °C. The Animal Care and Use Committee (MU-ACUC) of Mansoura University, Egypt, has approved all procedures involving animals with code number: MU-ACUC (PHARM.R.24.09.40). Throughout the experiment, medications and extracts were given orally in doses of 1 mL/100 g of the rat’s body weight after being solubilized in carboxymethyl cellulose (CMC) in order to create a suspension formula.

In this test, rats were divided into six groups, with each group containing 5 animals.

Group-1: The negative control group was given a vehicle, i.e., a 1% aqueous solution of CMC (10 mL/kg).

Group-2: Positive control (carrageenan-induced oedema, Sigma (St. Louis, MO, USA)) without any treatment.

Group-3: Were given the standard drug Ibuprofen at the dose of 10 mg/kg.

Group-4: Seedcake extract of *Lactuca sativa* was administered at a 2 g/kg dose.

Group-5: Compound (**1**) is administered at a dose of 20 mg/kg.

Group-6: Compound (**1**) at a dose of 40 mg/kg.

Group-7: Combination of standard and compound (**1**) at doses of (10 + 20) mg/kg, respectively.

After one hour, rats’ right hind paws were subcutaneously injected with 100 μL of carrageenan (1% in normal saline) to cause acute inflammation. Using a digital vernier calliper, the swelling of the paw that had been injected with carrageenan was measured prior to injection as well as 1, 2, 3, 4, and 5 h following the treatment’s induction. The percentage inhibition of oedema in the test animals treated with the extract compared to the carrageenan control group was used to calculate the anti-inflammatory activity. The formula %I = (dt/dc) × 100 was used to calculate the percentage inhibition of inflammation, where “dt” represents the difference in paw volume in the drug-treated group and “dc” represents the difference in paw volume relative to the control group. Moreover, “I” represents the suppression of inflammation [29].

##### Histopathological Assessment of Skin Tissue

Five hours following the carrageenan injection, tissue specimen samples from the paw skin tissue of every group under study were removed for histological examination. Prior to being submerged in paraffin blocks, they were fixed in a 10% formalin solution and stained with hematoxylin-eosin. Using a microtome, paraffin sections with a thickness of 5 µm were cut, and hematoxylin and eosin were frequently applied to the sections. Microscopically, these tissue slices were inspected. Consequently, the sections were examined under a light microscope; on the other hand, photos taken with a digital camera (Canon Power-shot D10) recorded the progression of paw oedema [30].

##### Determination of the Levels of Inflammatory Mediators and Oxidative Stress Parameters in Rat Paw Tissue Homogenate

Rats were slaughtered, and their paw tissues were gathered and weighed five hours after the carrageenan was administered. The tissues underwent processing after being fixed in 10% formalin. Rat paw tissue was homogenized, and its oxidative stress indices (SOD, CAT, and GPx) as well as the levels of inflammatory mediators (IL-1β, TNF-α, and PGE2) were measured [31].

#### 4.7.5. Anti-Inflammatory Activity (COX-1 and COX-2) Inhibition Assay

With some modifications, anti-inflammatory activity was carried out following Smith et al. 1998 [39], and ibuprofen served as the benchmark medication.

### 4.8. Molecular Docking of Compound *(**1**)* against COX-1 and COX-2

#### 4.8.1. Protein Preparation

The protein structures of COX-1 and COX-2 were acquired from their respective sources, Uniport, and are identified by the UniProt IDs A0A1B0RKU9 and P00406, respectively. The CB-Dock2 server [40] was utilized to estimate the active sites of these proteins. Using AutoDock Tools 1.5.7 [41], additional processing was performed on the produced protein structures, including the addition of polar hydrogen atoms, the assignment of Gasteiger charges, and the merger of non-polar hydrogen atoms. Molecular docking simulations were performed using the generated PDBQT files as input.

#### 4.8.2. Ligand Preparation

The active material that is extracted from the ligand molecules was obtained in their corresponding SDF formats from the PubChem database. Following this, they were minimized using the Conjugate Gradients technique and the Force Field (MMFF94) in the Avogadro 1.2.0 software [42]. The reduced ligand configurations were transformed into a PDBQT file that was compatible with AutoDock Vina [43], which was used to run molecular docking simulations. BIOVIA Discovery Studio 2020 [BIOVIA 2020] was used to visualize and analyze the docking data.

### 4.9. In Silico ADMET Prediction for Compound *(**1**)*

Compound **1**’s SMILES codes were entered into the ADMETlab 2.0 web server [44] to forecast the pharmacokinetic characteristics, such as absorption, distribution, metabolism, and excretion.

### 4.10. Statistical Analysis

The statistical analyses have been conducted using GraphPad Prism version 10.3.1 software. The data are presented as means ± standard deviations. Every determination was made three times, after which averages were estimated. The significance of the difference between means was determined by one-way ANOVA followed by Dunnett’s post hoc test, and the *p* < 0.05 values were considered significant.

## Figures and Tables

**Figure 1 ijms-25-11077-f001:**
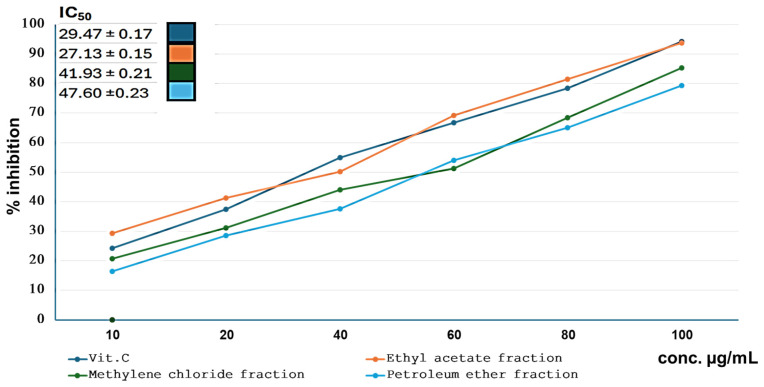
IC_50_ of different fractions of lettuce seedcake extract in ABTS assay using ascorbic acid as standard.

**Figure 2 ijms-25-11077-f002:**
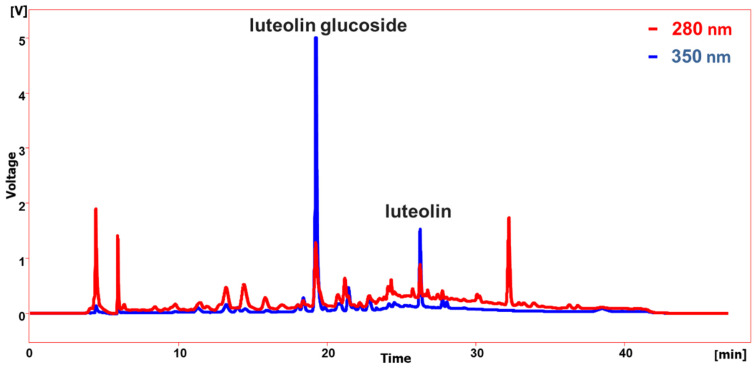
HPLC of the major compounds determined in lettuce seed cake extract. Compounds were monitored using a photo diode array (PDA) detector at 280, 350 nm.

**Figure 3 ijms-25-11077-f003:**
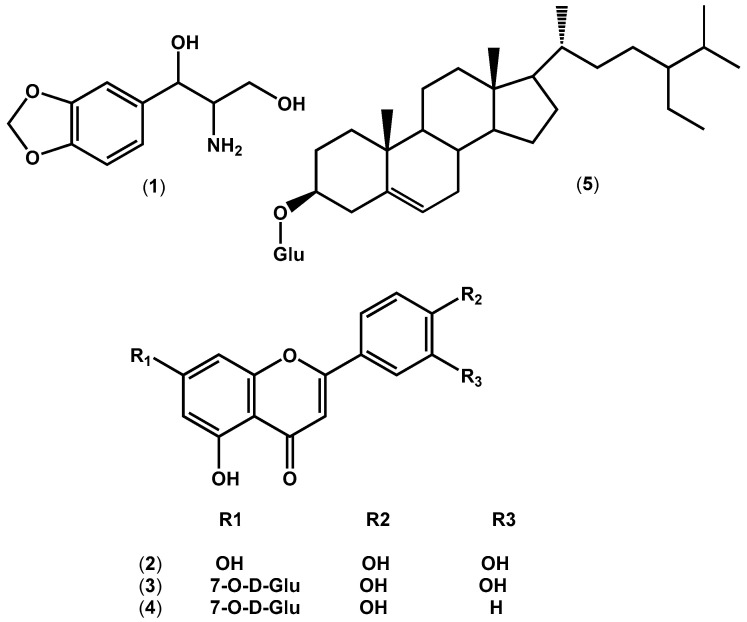
Structures of the isolated compounds from *Lactuca sativa* seedcake active fractions: 1,3-propanediol-2-amino-1-(3′,4′-methylenedioxyphenyl) (**1**), luteolin (**2**), luteolin 7-O-*β*-D glucoside (**3**), apigenin 7-O-*β*-D glucoside (**4**), and *β*-sitosterol 3-O-*β*-D-glucoside (**5**).

**Figure 4 ijms-25-11077-f004:**
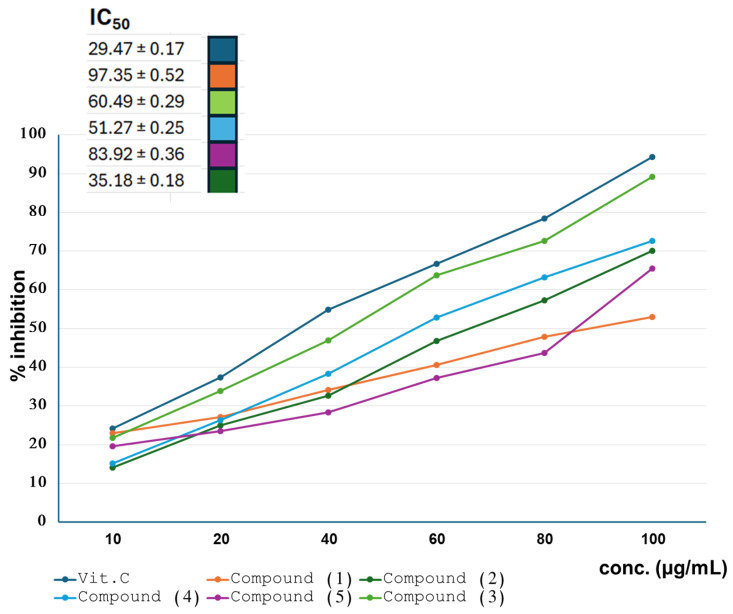
IC_50_ of isolated compounds from lettuce seedcake in ABTS assay using ascorbic acid as standard.

**Figure 5 ijms-25-11077-f005:**
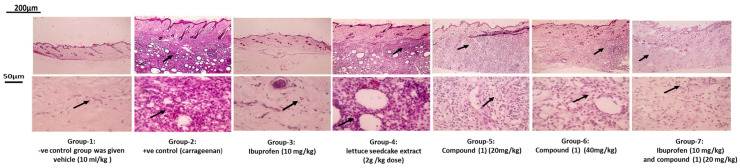
Microscopic pictures of H&E-stained skin sections. Magnifications X: 40 bar 200 are represented in the first column, thin black arrows refer to dermal inflammation and X: 400 bar 50 are represented in the second column where thin black arrows indicate leukocytic cells infiltration in dermis.

**Figure 6 ijms-25-11077-f006:**
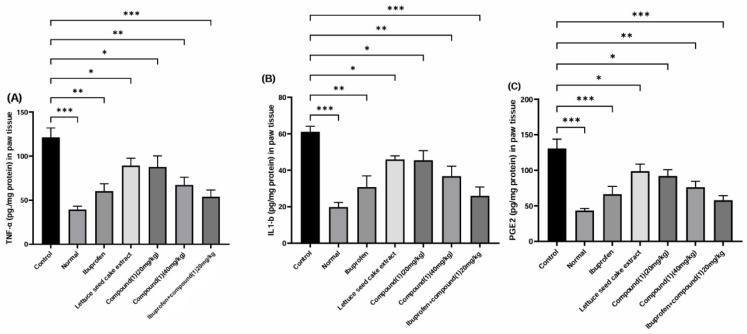
Effect of treatments on TNF-α (**A**), IL-1β (**B**), and PGE2 (**C**) in carrageenan-induced oedema in rat hind paws. Values represent mean ± SD for each group (*n* = 6). * *p* < 0.05, ** *p* < 0.01, *** *p* < 0.001.

**Figure 7 ijms-25-11077-f007:**
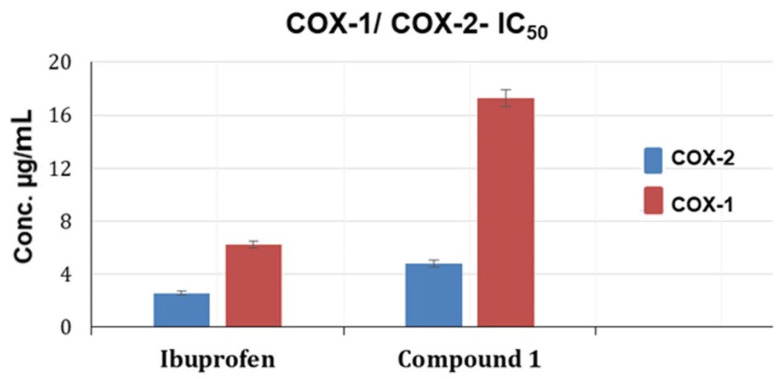
IC_50_ of compound **1** and Ibuprofen standard for COX-1 and COX-2 inhibition assay.

**Figure 8 ijms-25-11077-f008:**
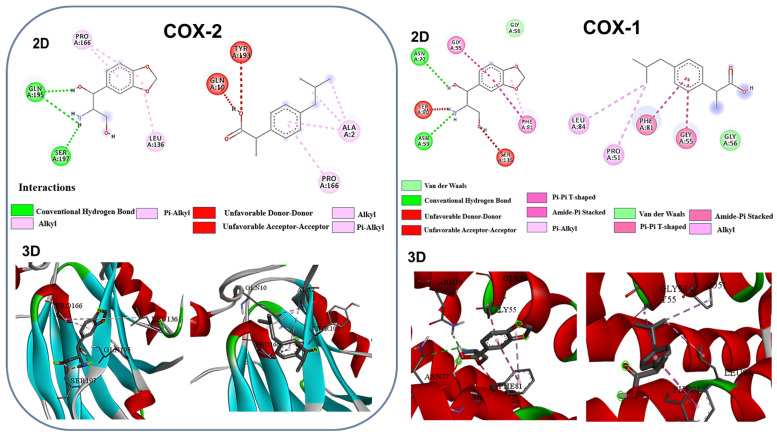
The 3D and 2D representations of compound **1** and ibuprofen interacting with the COX-1 and COX-2 enzymes are shown. The 2D diagrams illustrate color-coded interactions between the ligand and specific amino acid residues, while the 3D models display the ligand within each enzyme’s binding pocket, highlighting key structural features. In 3D model of protein (red: alpha helices; cyan: beta sheets; green: loops) complexed with drug (stick model).

**Table 1 ijms-25-11077-t001:** Antimicrobial activity index (mean ± SE) of compounds isolated from lettuce seedcake extract (1 mg/mL).

Compound No.	*E. coli*		*P. aeruginosa*		*S. aureus*		*C. Albicans*	
	Diameter of Inhibition Zone (mm)	%Activity Index	Diameter of Inhibition Zone (mm)	%Activity Index	Diameter of Inhibition Zone (mm)	%Activity Index	Diameter of Inhibition Zone (mm)	%Activity Index
**1**	NA *	----	3.8 ± 0.2	16.5	6.9 ± 0.1	28.9	NA *	----
**2**	9.8 ± 0.2	37.8	12.9 ± 0.1	56.6	17 ± 0.1	71.7	10.7 ± 0.15	40.5
**3**	8.8 ± 0.15	33.8	11.77 ± 0.25	51.5	14.83 ± 0.15	61.9	9.87 ± 0.15	36.7
**4**	6.8 ± 0.15	26.1	9.8 ± 0.15	42.8	11.87 ± 0.15	50	5.95 ± 0.07	22.1
**5**	5.1 ± 0.1	19.6	7.9 ± 0.1	34.5	10.83 ± 0.2	49.6	4.9 ± 0.1	18.2
**Ciprofloxacin**	26 ± 0.3	100	22.87 ± 0.15	100	23.9 ± 0.1	100	NA *	----
**Clotrimazole**	NA *	----	NA *	----	----	----	26.9 ± 0.1	100

* NA = No Activity.

**Table 2 ijms-25-11077-t002:** Cytotoxic activity of isolated compounds from lettuce seedcake against (HepG2) and (MCF-7) cancer cell lines and compared with its effect on normal cell lines.

Compound No.	In Vitro Cytotoxicity IC_50_ (µg/mL)			Selectivity Index	
	HepG-2	MCF-7	WI-38	HepG2	MCF-7
**1**	82.75 ± 4.1	86.85 ± 3.9	61.71 ± 3.4	0.75	0.71
**2**	19.50 ± 1.3	11.63 ± 0.9	59.33 ± 3.2	3	5.1
**3**	24.83 ± 1.9	17.79 ± 1.3	49.64 ± 2.7	2	2.8
**4**	43.35 ± 2.5	39.79 ± 2.2	84.30 ± 4.3	1.9	2.1
**5**	63.59 ± 3.4	52.76 ± 2.9	34.77 ± 2.1	0.54	0.66
**Doxorubicin**	4.50 ± 0.2	4.17 ± 0.2	6.72 ± 0.5	1.5	1.6

**Table 3 ijms-25-11077-t003:** Treatments effect on carrageenan-induced paw oedema.

Treatment	Dose (mg/kg)	Increase in Paw Oedema (mL) and % Inhibition (%I)				
		1 h	2 h	3 h	4 h	5 h
Control	-	1.47± 0.08	3.33 ± 0.05	3.95 ± 0.04	3.99 ± 0.40	4.13 ± 0.46
Ibuprofen	10	0.69 ± 0.16 (15.62)	1.57 ± 0.21 ** (25.62)	1.82 ± 0.38 *** (28.5)	1.49 ± 0.47 *** (62.66)	1.20 ± 0.37 ** (70.9)
Lettuce seed cake extract	2000	1.29 ± 0.11 (5.87)	2.20 ± 0.44 ** (18)	2.59 ± 0.19 *** (19.57)	2.49 ± 0.13 *** (37.5)	2.42 ± 0.10 ** (41.40)
Compound (**1**)	20	1.05 ± 0.20 (10.25)	1.81 ± 0.44 ** (23.51)	2.26 ± 0.28 *** (23.77)	2.18 ± 0.74 *** (45.36)	1.75 ± 0.46 ** (57.63)
Compound (**1**)	40	0.77 ± 0.28 (16.32)	1.59 ± 0.22 ** (27.07)	2.11 ± 0.29 *** (26.1)	1.78 ± 0.45 *** (55.39)	1.49 ± 0.49 ** (63.92)
Ibuprofen + Compound (**1**)	10 + 20	0.58 ± 0.17 (20)	1.30 ± 0.29 ** (31.28)	1.42 ± 0.57 *** (35.35)	1.12 ± 0.40 *** (71.93)	0.83 ± 0.34 ** (79.90)

Values are expressed as mean ± S.D. ** *p* < 0.01, and *** *p* < 0.001, compared to the vehicle control group. Differences between groups were analyzed by analysis of variance (one-way ANOVA) followed by Dunnett’s test. The standard anti-inflammatory drug ibuprofen (10 mg/kg) also significantly decreased paw oedema (*** *p* < 0.001) after 3rd and 5th h of carrageenan administration.

**Table 4 ijms-25-11077-t004:** Effects of treatments and ibuprofen on CAT, SOD, and GPx activities in carrageenan-induced paw odema.

Treatment	GPx (U/gm Tissue)	CAT (U/gm Tissue)	SOD (U/gm Tissue)
Normal	118.55 ± 10.25 ***	2.83 ± 0.11 ***	189.05 ± 13.79 ***
Control	52.85 ± 7.99	0.87 ± 0.13	76.55 ± 10.68
Ibuprofen	93.35 ± 8.84 **	2.20 ± 0.20 ***	142.1 ± 14.71 **
Lettuce seed cake extract	77.05 ± 3.89 *	1.45 ± 0.08 *	116.65 ± 1.77 *
Compound (**1**) (20 mg/kg)	77.65 ± 3.18 *	1.50 ± 0.19 *	118.35 ± 1.06 *
Compound (**1**) (40 mg/kg)	86.75 ± 6.01 *	2.04 ± 0.26 **	135.05 ± 11.67 *
Combination (Ibuprofen + compound (**1**) (20 mg/kg))	99.85 ± 6.72 **	2.36 ± 0.26 ***	160.85 ± 15.06 **

Values are expressed as mean ± S.D. * *p* < 0.05, ** *p* < 0.01, and *** *p* < 0.001, compared to the vehicle control group. Differences between groups were analyzed by analysis of variance (one-way ANOVA) followed by Dunnett’s test.

**Table 5 ijms-25-11077-t005:** ∆G (kcal/mol) for each compound with tested proteins (COX1 and COX2).

Compound	COX-1	COX-2
Ibuprofen	−5.4	−4.8
Compound **1**	−6.6	−5.4

## Data Availability

Data are contained within the article and Appendix A.

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
