# Peer review of "Oilseed Cakes: A Promising Source of Antioxidant, and Anti-Inflammatory Agents—Insights from *Lactuca sativa"

_ijms, 2024, doi:10.3390/ijms252011077_

Round 1

Reviewer 1 Report

Comments and Suggestions for Authors

In the manuscript by Majed et al. entitled »Oilseed Cakes: A Promising Source of Antioxidant, and Anti-Inflammatory Agents—Insights from Lactuca sativa«, the authors investigated the antioxidant and antibacterial properties of methanolic extracts, obtained from oilseed cakes of Lactuca sativa (lettuce), Nigella sativa (black seed), Eruca sativa (rocket) and Linum usitatissimum (linseed). The methanolic extract from lettuce was selected for further investigation as it was the most promising in terms of its antioxidant and antibacterial properties. Several compounds were isolated from the extract using high-performance liquid chromatography. The isolated compounds were tested on liver (HepG2) and breast cancer cell lines (MCF-7) and analysed for their antioxidant, antimicrobial and cytotoxic effects. In addition, the anti-inflammatory activity of lettuce seed cake extract and 1,3-propanediol-2-amino-1-(3',4'-methylenedioxyphenyl) was evaluated in vivo using carrageenan-induced paw oedema model. It was shown that 1,3-propanediol-2-amino-1-(3',4'-methylenedioxyphenyl) and its combination with ibuprofen can substantially reduce paw oedema and restore antioxidant enzyme activity. These experimental results were supported by molecular docking calculations.

The subject of the study is interesting, although I personally do not believe that the proposed application of oilseed cakes of Lactuca sativa would be commercially viable. From a scientific point of view, the paper is correctly written and no major flaws have been identified. It appears that the study meets the recognised standards and is also original enough to be published in the International Journal of Molecular Sciences.

Nevertheless, some points in the study are not clearly formulated, so I propose to publish the study after a minor revision of the paper has taken place, taking into account my comments below:

1.) Line 387 states "... 200 gm of lettuce … ”. What unit is “gm”? If it is grammes, then the abbreviation is “g”.

2.) In line 394 it says "... 80% A, 20% B for 25 minutes; 80% A, 20% B for 30 minutes; ...". Obviously the elution was not isocratic, so the authors should clearly explain how the composition of the mobile phase changes with time. I assume that there is an error (lapsus calami) in the given elution gradient, because otherwise it would simply read "... 80% A, 20% B for 55 minutes; ...". Alternatively, the authors could present the gradient programme graphically (which would probably be the clearest way of presenting it).

3.) Line 455 reads: “The negative control group was given a vehicle (10 ml/kg)”. I am not an expert in drug administration (like probably most readers of the International Journal of Molecular Sciences) and do not know what means “vehicle”in this particular case. Please explain.

4.) There are some minor problems with the English language (e.g. line 351 “Oil seed cakes are a hidden treasures”).

Comments on the Quality of English Language

There are some minor problems with the use of English language.

Reviewer 2 Report

Comments and Suggestions for Authors

The paper “Oilseed Cakes: A Promising Source of Antioxidant, and AntiInflammatory Agents—Insights from Lactuca sativa” evaluates the properties of methanolic extracts of different oilseed cakes with the aim of exposing their potential as sources of bioactive compounds.

-          It is recommended to update the title in order to reflect the fact that other oilseed cakes were analyzed besides Lactuca sativa

2. Results and Discussion

-          The conducted examinations reveal that the extracts possess anti-oxidant and antimicrobial properties. A comparison of the reported outcomes with the results available for other extracts obtained from other types of oilseed cakes would improve the paper quality.

4. Materials and methods

-          It is recommended to add an initial sub-section dedicated to the reagents used for the experiments with indication about the providers, purity etc.

4.3. Extraction of seed oil cakes derived from different plant species

-          “For preliminary screening to obtain a methanolic extract of different oilseed cakes, 200 gm of lettuce, black seed, rockets, and flaxseed seed cakes were extracted using methanol solvent (2x 400 mL each).” – The extraction method should be added.

-          Please replace “gm” with “g” as measurement unit.

-          The methanolic extracts were directly used or they were first concentrated?

-          If the maceration was used as extraction method, how were the extracts separated from the mixture?

-          Why this particular method of extraction was used? Were other methods tested? With what results?

4.4. HPLC analysis of lettuce seed cake defatted extract

-          Please specify the solvents A and B used for the chromatographic analysis.

4.5. Bio-guided isolation of active constituents from Lactuca sativa seed cake active fractions

-          The verb is missing from the phrase “The resulting petroleum ether extract (590 g, extremely thick and oily), methylene chloride extract (11 g), and ethyl acetate extract (14 g).”

Comments on the Quality of English Language

Minor editing of English language required.
